


# Airborne radionuclides and heavy metals in High Arctic terrestrial environment as the indicators of sources and transfers of contamination

Edyta Łokas[1], Agata Zaborska[2], Ireneusz Sobota[3], Paweł Gaca[4], Andrew Milton[4], Paweł Kocurek[5], Anna Cwanek[1]

[1]Department of Nuclear Physical Chemistry, Institute of Nuclear Physics Polish Academy of Sciences, Kraków, 31-342, Poland
[2]Marine Chemistry and Biochemistry Department, Institute of Oceanology Polish Academy of Sciences, Sopot, 81-712, Poland
[3]Department of Hydrology and Water Management, Polar Research Centre, Nicholas Copernicus University, Toruń, 87-100, Poland
[4]Ocean and Earth Science, University of Southampton, National Oceanography Centre, European Way, Southampton, SO14 3ZH United Kingdom
[5]Research and Development Laboratory for Aerospace Materials, Rzeszow University of Technology, Rzeszow, 35-959, Poland

*Correspondence to*: Edyta Łokas (Edyta.Lokas@ifj.edu.pl)

**Abstract.** A survey of airborne radioactive isotopes ($^{137}$Cs, $^{238}$Pu, $^{239+240}$Pu, $^{241}$Am and $^{210}$Pb) and trace metals (Pb, Cu, Zn, Cd, Fe, Al) in tundra soils and cryoconite hole material sampled from several locations in the Kaffiøyra region revealed high spatial concentration variability of the analysed samples. Lithogenic radionuclides ($^{230}$Th, $^{232}$Th, $^{234}$U, $^{238}$U) show less variability than the airborne radionuclides because their activity concentrations are controlled only by mixing of weathered material derived from different bedrock.

Activity ratios of the artificial radionuclides differ in most cryoconite samples from global fallout signatures. The contribution of radionuclides from other sources might be enhanced by non-continuous exposure of cryoconite to atmospheric deposition. We assumed that the main source of Pu, which is visible only in cryoconite samples, are derived from nuclear tests and non-exploded weapons-grade material. Approximately one third of the total observed Pu activity concentration is $^{238}$Pu originating from a SNAP9A satellite re-entry and subsequent injection of nuclear debris from stratosphere into troposphere. In samples from Waldemarbreen we observed the effect of glacial morphology on effective trapping and storing of airborne radionuclides. The differences in the concentrations of radionuclides between sampling points and differences in the elevation gradient from terminus towards icefall may reflect the homogenous topography of the glacier tongue. The trace metal concentrations in soils were typical or slightly higher than concentrations characteristic for natural background concentrations and the $^{206}$Pb/$^{207}$Pb ratio also was close to the natural ratio for parent rocks. Conversely, trace metal concentrations in cryoconite samples (Pb and Cd) were higher than in soil samples and definitely exceeded natural values.



# 1 Introduction

The occurrence of radioactive elements in the environment is related to both natural and anthropogenic factors. The artificial radionuclides ($^{137}$Cs, $^{238,239,240}$Pu, $^{241}$Am) were released into the environment due to various human activities mainly in the second half of the 20th century. They were produced through nuclear fission and neutron activation processes. Main
sources of anthropogenic radionuclides in Northern Hemisphere are: (i) nuclear weapon tests, atmospheric explosions in Novaya Zemlya, Semipalatinsk, Nevada etc.; (ii) nuclear accidents (Kyshtym-1957, Lake Karchay-1968; Tomsk-1993; Chernobyl-1986; Fukushima-2011); (iii) disintegration of satellites (SNAP9A-1964, Cosmos 958-1978). Relatively small but constant releases of artificial radioisotopes are also associated with reprocessing of spent nuclear fuel and energy generation in nuclear power stations. Studies of the presence of anthropogenic radionuclides in the environment have entered
a new era with the renaissance of nuclear energy and associated fuel reprocessing. The renewed interest in environmental radioactivity is closely connected with potential threats to national security and non-proliferation of nuclear material. Additionally, the presence of these artificial radionuclides in the environment may potentially be harmful for humans and for the ecosystems as it contributes to the overall dose of ionising radiation received by organisms as well as introduces harmful chemicals previously not present in ecosystems with the potential to accumulate at various levels of a trophic chain.

The other contaminant group that is recognized as a pollutant of the cryosphere are heavy metals. Heavy metals are natural elements of the Earth's crust. Since the beginning of the industrial era (from~1850) loading of metals to the environment due to human activities has increased nearly 10 times (Nriagu, 1996). The emission of heavy metals increased substantially after World War II and is still increasing in some countries (Asian sector). The main anthropogenic heavy metal sources include industry, mining, agriculture, fuel burning, waste disposal and transportation (Pacyna and Pacyna, 2001).
Heavy metals pollution usually occurs locally but can be also transported globally by air mass circulation, rivers, oceanic currents, etc. Heavy metals have been found in Asian and European glaciers (Barbante et al., 2004; Aizen et al., 2009; Eyrikh et al., 2017; Baccollo et al., 2017) and even in remote Arctic and Antarctic areas (Hur et al., 2007; Singh et al., 2013). As they melt, glaciers release metals to streams that may pollute both vegetation and drinking water sources (Dong et al., 2017).

The highest concentrations of pollutants on glaciers are most likely stored in cryoconite granules and micro-fauna (Segawa et al., 2013; Łokas et al., 2016; Baccolo et al., 2017; Zawierucha et al., 2018). Cryoconite granules are aggregates of mineral and organic components where it is possible to identify biological consortia composed of archaea, algae, cyanobacteria, fungi and heterotrophic bacteria (Cook et al., 2015; Simon et al., 2002; Takeuchi et al., 2001a,b; Hodson et al., 2008). Cyanobacteria play a crucial role in formation of such aggregates. They produce extracellular polymeric
substances whose adhesive properties enhance the accumulation of mineral dust and microorganisms. Among the many effects played by cryoconite in glacial environments, their influence on ice albedo is extremely relevant. Given their dark color, cryoconite is able to significantly increase the absorption of solar light by glaciers, enhancing and speeding up melting processes (Simon et al., 2002; Takeuchi et al., 2001a,b; Hodson et al., 2008). Cryoconites are usually found in spatially





confined holes, produced by locally increased ice melting caused by the dark color of cryoconite granules themselves. They usually form on the ablation zone of glaciers and constitute nutrient-rich pools with diverse biota within the supraglacial zone (Wharton et al., 1985; Cook et al., 2016). Glaciers all over the world are projected to lose 80 % of their volume by the end of 2100 and some of them are expected to disappear within decades at current climatic conditions (Radic et al., 2014).

Cryoconite granules containing high concentrations of contaminants and can persist on glacier surfaces for decades (Łokas et al., 2014), nevertheless, as the glacier ice disappears, they deposit their contents on the freshly exposed areas. The discovery of hot spots of radioactivity in the proglacial zones of Spitsbergen glaciers (Łokas et al., 2014; Łokas et al., 2017b) provides evidence that cryoconite-derived contaminants may have accumulated in depressions of the glacier forefield, together with fines redistributed by slope wash.

A critical feature of a cryoconite is its accumulation capability. Given these properties, cryoconite samples are among the best candidates to monitor the spreading of artificial radionuclides and heavy metals into their surrounding environments. Determination of Pu isotopes, $^{241}$Am and $^{137}$Cs in cryoconite samples is of great interest for several reasons; as a radioecological monitoring tool because of the development of nuclear technologies for military purposes (India, Pakistan and potentially Iran) and the rising number (and risk) of nuclear events in civilian nuclear installations as well as the growing

risk of nuclear terrorism and undeclared nuclear activities.

The main objective of this study is to evaluate the concentrations of artificial ($^{137}$Cs, $^{238,239+240}$Pu, $^{241}$Am) and natural ($^{210}$Pb, $^{230}$Th, $^{232}$Th, $^{234}$U, $^{238}$U) radionuclides and trace metals (Pb, Cu, Zn, Cd, Fe, Al) of cryoconite holes and tundra soils sampled in several locations in the Kaffiøyra region, Svalbard (Fig. 1A) and to identify and constrain the different sources of contamination based on the isotopic ratios of radionuclides ($^{238}$Pu/$^{239+240}$Pu, $^{241}$Am/$^{239+240}$Pu, $^{239+240}$Pu/$^{137}$Cs, $^{240}$Pu/$^{239}$Pu,

$^{206}$Pb/$^{207}$Pb and $^{208}$Pb/$^{206}$Pb).

## 2 Data and methods

### 2.1 Study area

The Kaffiøyra region, located in the north-western Spitsbergen, the adjoining Aavatsmarkbreen (75 km$^2$) and the Dahlbreen (132 km$^2$) together comprise an area of about 310 km$^2$ (Sobota et al., 2016). It is a coastal lowland at Forlandsundet (Forland Sound). It accounts for 12 % of the area of Oscar II Land. Along with Kaffiøyra, which is 14 km

long and 4 km wide, seven land glaciers are located in this region. The main pedogenetic processes in the Kaffiøyra region consist of the accumulation of organic material, gleying and decarbonation. Cryogenic processes, which overlap the pedogenetic processes, also play an important role in the genesis of the soils there. The substantial moisture content of the surface layers and stagnant waters result in a shallow permafrost (Plichta, 2005). In the years 1996–2015, the greatest

average active layer thickness of permafrost (215 cm) was recorded at a location in the moraine. A much lower mean value was recorded on the beach (126 cm). An active layer in the tundra reached on average a depth of 150 cm (Sobota et al. 2018). Climate research regarding the Kaffiøyra region has been conducted since 1975. The increase in air temperature noted





during the study confirms the general trend observed in Svalbard and in many areas of the Arctic (Nørdli et al., 2014). In the years 1997–2016 an average air temperature during the summer season in this region was 5.4°C.

Glaciers of this region are polythermal (Sobota, 2009, 2011). From the time of their maximum extent in the late 19[th] and early 20[th] centuries to 2015, the total area of this region's valley glaciers has decreased on average by about 43 %

(Sobota et al. 2016). The Waldemarbreen is an alpine-type glacier running down the valley towards the Kaffiøyra. It contains one accumulation zone and a glacier tongue in the valley. The area of the glacier is 2.40 km² (Sobota, 2017). It borders the Prins Heinrichfjella ridge in the north and east (500–770 m above sea level), and Gråfjellet (300–350 m) in the south. The glacier consists of two distinct parts separated by a medial moraine. The mean annual mass balance of the Waldemarbreen in 1996–2015 was -0.72 m w.e. (Sobota et al. 2016) and in 2014 ablation of glacier also was very significant. Starting from

1909, the glacier has been receding by 8 m a⁻¹. Between years 1995–2009 the recession rate accelerated to 10 m a⁻¹, and between 2000–2009 reached 11 m a⁻¹ (Sobota and Lankauf, 2010). From the time of the maximum advance to 2015 the Waldemarbreen area decreased by 31.4 % and the annual average recession rate was 8 ma⁻¹ (Sobota et al., 2016) (Fig. 1B).

## 2.2 Field campaign

The locations of sampling sites are presented in Fig. 1A. Soil samples were collected in 2009 (coded: S01-S06).

Sampling sites represent accumulations of typical tundra samples. Generally, this area is predominantly covered with dry moss and lichen tundra with communities of Saxifraga opositifolia, Salix polaris – Sanionia uncinata, Luzula arcuata ssp. confusa – Cetrariella delisei – Scorpidium revolvens (Sobota et al. 2018). Soil profiles S01-S06 were collected at increasing distances from the front of Waldemarbreen (soil profile S06 was located on the flat outwash fan at the foreland of the moraine ridge of Aavatsmarkbreen). The profile depths varied from 12 to 17 cm. All these profiles were covered by a layer

of organic material. All soil cores were divided into 1−2 cm sub-samples and dried at 105° C to a constant weight, passed through a 2 mm sieve and then prepared for radionuclide analysis. Additionally, twelve cryoconite samples (coded: 2-13) were collected in the Waldemarbreen in August-September 2014 with disposable plastic Pasteur pipettes from the bottom of randomly selected cryoconite holes and transferred to 15 cm³ plastic test tubes. Cryoconite samples in this study were taken from different altitudes, from cryoconite holes characterised by different depths and surface areas (Tab. S1).

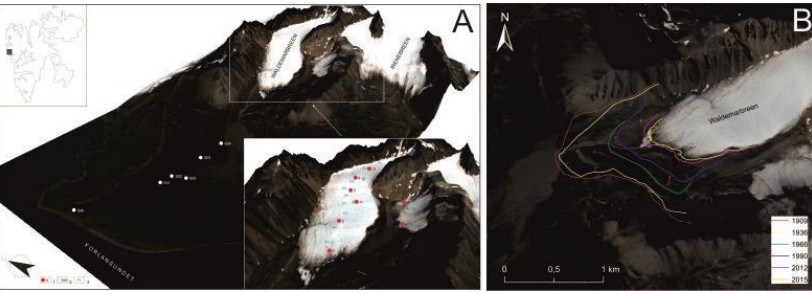


Figure 1. A. Location of the study area. Profile S01-S06 were collected at increasing distances from the front of Waldemarbreen and cryoconite samples (2-13) from different altitudes of this glacier. B. Map of recession of the Waldemarbreen in 1909-2017.



## 2.3 Methods

### 2.3.1 Radionuclide analyses

Gamma emitting radionuclides ($^{137}$Cs and $^{210}$Pb) were measured by planar HPGe (high-purity germanium) detector. The activities of $^{137}$Cs were determined using its emission peak at 662 keV and for $^{210}$Pb the 46.6 keV. Corrections were
made for the effect of self-absorption of low-energy g-rays (46.6 keV) within the sample even though these corrections were insignificant because of vary low sample masses (from 2 to 12 g dry masses). The activities of $^{238}$Pu, $^{239+240}$Pu, $^{241}$Am, $^{234,238}$U and $^{230,232}$Th were determined for 0.5–1.15 g cryoconite samples and 5 g dried soil samples. Organic matter was decomposed by a muffle oven ignition at 600° C for 6 hours. The samples were then dissolved using concentrated HF, HNO$_3$, HCl, and a small addition of H$_3$BO$_3$. Details of the sequential radiochemical procedure used for determination of
$^{238}$Pu, $^{239+240}$Pu, $^{241}$Am, $^{234,238}$U and $^{230,232}$Th are described in previous publications (Łokas et al., 2010; Łokas et al., 2018). The $^{240}$Pu/$^{239}$Pu ratio was determined using a Thermo Fisher Scientific Neptune MC-ICP-MS (analytical method based on Taylor et al. 2001; Łokas et al., 2018). The Environmental Radioactivity Laboratory of the Henryk Niewodniczanski Institute of Nuclear Physics holds an ISO17025 accreditation for gamma-spectrometric measurements and Pu analyses by alpha-particles spectrometry.

### 2.3.2 Heavy metal analyses

Selected heavy metals that are believed to be important anthropogenic contaminants (Pb, Cu, Zn, Fe, Al) were measured with a flame atomic absorption spectrometer (AAS Shimadzu 6800) using deuterium background correction. Cd and the Pb stable isotopic ratios were measured on a Perkin-Elmer Sciex ELAN 9000 ICP-MS. The exact measurement procedure and quality assurance is described in detail in Zaborska et al. (2017). The metal enrichment factors (EF) were
calculated according to the formula presented in Zaborska et al. (2017). The mean background metal concentrations were taken from literature for the region of Hornsund. The fraction of anthropogenic Pb (Pb$_{ant}$) was calculated according to formula presented in Zaborska et al. (2017). The excess $^{206}$Pb/$^{207}$Pb ($^{206}$Pb/$^{207}$Pb$_{excess}$) ratio calculation was used to find a source of Pb pollution (Farmer et al., 1996; Bindler et al., 2001).

*Statistical analyses*

The regression analysis was performed to find correlations between particular radionuclide and metal concentrations. The correlations characterized by r>0.5 are assumed to show moderate correlation, the r>0.7 is assumed to show strong correlation. Nonparametric Kruscal-Wallis test was used to find a difference between radionuclide and metal concentrations in soil and cryoconite samples. All statistical calculations were performed using STATISTICA 10.0 licensed program (StatSoft, Inc. Team. 2011).

## 3 Results

### 3.1 Radionuclide contents and their activity ratios in tundra soil profiles



Activity concentrations and inventories of anthropogenic radionuclides ($^{137}$Cs, $^{238}$Pu, $^{239+240}$Pu, $^{241}$Am) for tundra soil profiles are presented in Tab. S2 and in Fig. 2A. Radionuclide inventory is understood here as the activity of a given radionuclide contained in the soil column at unit surface area. Most of the profiles have concordant depth distributions of all artificial radionuclide activity with distinct peaks occurring in the surface or the first sub-surface layers. The highest activity concentrations of anthropogenic radionuclides are observed in profile S04 at 1 cm depth where significant amounts of organic matter were observed (21 % LOI). Profile S02 shows no detectable $^{137}$Cs with a minimal detectable activity (MDC) range of 0.1 to 3 Bq kg$^{-1}$. Activities of other radionuclides in this profile were not determined.

The determined activity concentrations of $^{137}$Cs, $^{238}$Pu, $^{239+240}$Pu and $^{241}$Am varied between 7±2 to 65±7 Bq/kg, 0.03±0.01 to 0.09±0.01 Bq/kg, 0.08±0.01 to 2.13±0.16 Bq/kg and 0.09±0.01 to 0.90±0.06 Bq/kg, respectively. In most of the samples the activity concentrations of $^{238}$Pu are close to the detection limit, which is why the activity ratios of $^{238}$Pu/$^{239+240}$Pu were calculated for only 3 samples. These $^{238}$Pu/$^{239+240}$Pu activity ratios varied between 0.029±0.010 and 0.05±0.02 with the mean value of 0.039±0.010. The $^{240}$Pu/$^{239}$Pu atomic ratios found in tundra profiles varied from 0.168±0.001 to 0.191±0.005 with an average of 0.179±0.006 (Łokas et al., 2017a). The $^{239+240}$Pu/$^{137}$Cs activity ratios range between 0.025±0.003 to 0.059±0.009 with the mean value of 0.042±0.010. The $^{241}$Am/$^{239+240}$Pu activity ratios range between 0.28±0.04 and 0.55±0.11 with the mean value of 0.46±0.07. Artificial radionuclide inventories evaluated for soils vary between 150±40 to 670±100 Bq m$^{-2}$ for $^{137}$Cs, from 0.44±0.15 to 0.96±0.02 Bq m$^{-2}$ for $^{238}$Pu, from 6.8±0.6 to 28±2 Bq m$^{-2}$ for $^{239+240}$Pu and from 1.9±0.2 to 14±3 Bq m$^{-2}$ for $^{241}$Am. The average values of the radionuclide inventories calculated for all of these profiles are: 0.66±0.14 Bq m$^{-2}$ for $^{238}$Pu, 20±2 Bq m$^{-2}$ for $^{239+240}$Pu, 9±1 Bq m$^{-2}$ for $^{241}$Am and 450±70 Bq m$^{-2}$ for $^{137}$Cs.

The measurement results for natural radioisotopes ($^{210}$Pb, $^{234,238}$U, $^{230,232}$Th) for tundra soil profiles are presented in Figure 2A and B. Activity concentrations range from 15±1 to 173±4 Bq kg$^{-1}$ for total $^{210}$Pb, 13±1 to 28±2 Bq kg$^{-1}$ for $^{234}$U, 13±1 to 30±2 Bq kg$^{-1}$ for $^{238}$U, 9±1 to 22±2 Bq kg$^{-1}$ for $^{230}$Th and 8±1 to 42±3 Bq kg$^{-1}$ for $^{232}$Th. The highest activity concentration for $^{210}$Pb was found in the surface layer of profile S04. Activity ratios of $^{234}$U/$^{238}$U varied between 0.9 ± 0.1 to 1.2 ± 0.2 (Tab. S3).

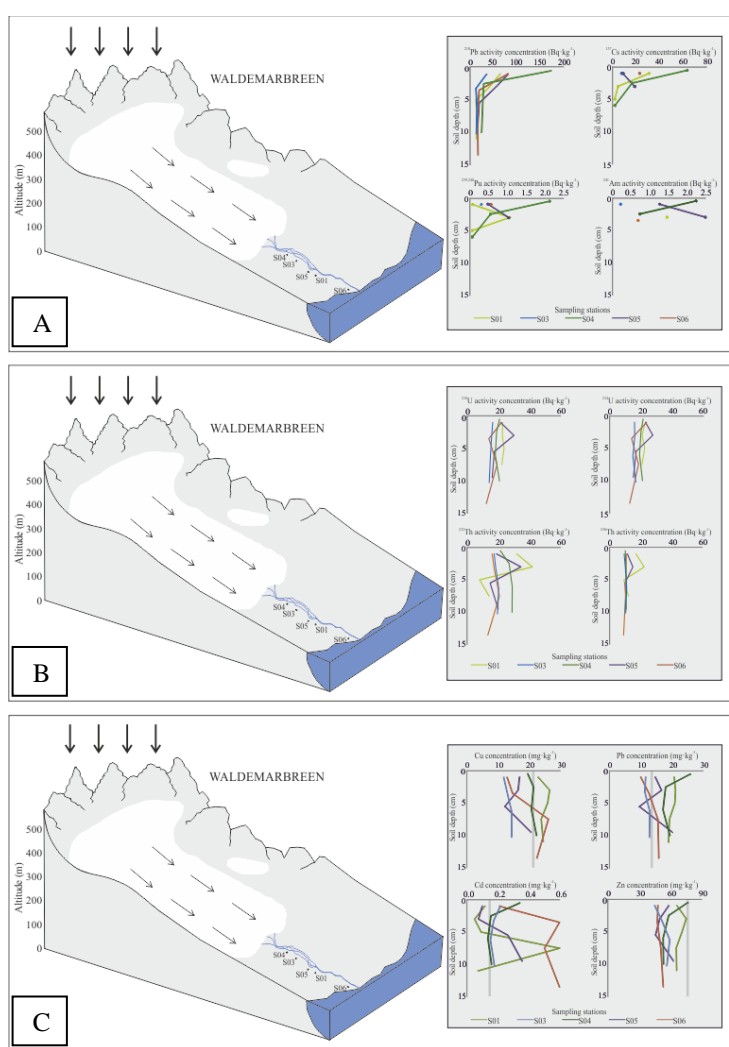

Figure 2. The model of Waldemarbreen and depth distribution of atmospheric radionuclides ([210]Pb, [137]Cs, [239,240]Pu, [241]Am) (A), depth distribution of natural radionuclides ([234, 238]U, [230,232]Th) (B) and depth distribution of selected heavy metals (Cu, Pb, Cd, Zn) (C) in tundra soils.

## 3.2 Radionuclide contents and their activity ratios in cryoconite samples

The results of activity concentrations of anthropogenic radionuclides analysis ([137]Cs, [238]Pu, [239+240]Pu and [241]Am) for all the cryoconite samples are presented in Table S4 and Fig. 3. Activity concentrations range from: $13\pm3$ to $2000\pm300$ Bq kg[-1] for [137]Cs, $0.08\pm0.02$ to $2.1\pm0.2$ Bq kg[-1] for [238]Pu, $0.09\pm0.02$ to $43\pm3$ Bq kg[-1] for [239+240]Pu and $0.25\pm0.06$ to $25\pm2$ Bq kg[-1] for [241]Am. All of these values are significantly higher than those observed in soil samples collected near this glacier.



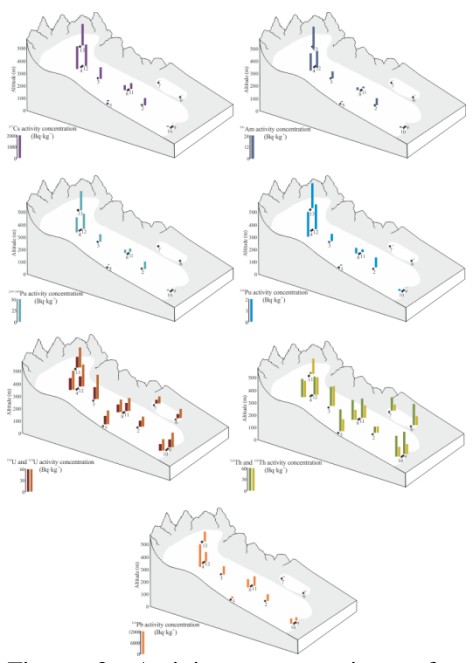

Figure 3. Activity concentrations of anthropogenic ($^{137}$Cs, $^{239,240}$Pu, $^{241}$Am) and natural ($^{234,238}$U, $^{230,232}$Th, $^{210}$Pb) radionuclides in cryoconite samples from Waldemarbreen.

The activity ratios of $^{238}$Pu/$^{239+240}$Pu, $^{239+240}$Pu/$^{137}$Cs and $^{241}$Am/$^{239+240}$Pu, as well as atomic ratios of $^{240}$Pu/$^{239}$Pu, were calculated to potentially distinguish the sources of these radionuclides in cryoconite samples and are presented in Table S4. The $^{238}$Pu/$^{239+240}$Pu activity ratios varied between 0.031±0.006 (sample 11) and 0.062±0.008 (samples 4 and 12). The mean value for activity ratios of $^{238}$Pu/$^{239+240}$Pu is 0.050±0.009. The analysis of atomic ratios of $^{240}$Pu/$^{239}$Pu provides important information which allows a more precise identification of the origin of Pu isotopes in environmental samples. $^{240}$Pu/$^{239}$Pu

atomic ratios found in cryoconite samples varied from 0.121±0.039 (sample 9) to 0.196±0.019 (sample 5) with an average of 0.147±0.009. The $^{239+240}$Pu/$^{137}$Cs activity ratios in the cryoconite samples range between 0.007±0.003 to 0.038±0.008 with the mean value of 0.020±0.004. The $^{241}$Am/$^{239+240}$Pu activity ratios range between 0.24±0.05 and 0.65±0.12 with the mean value of 0.47±0.07. Data on natural radioisotopes ($^{210}$Pb, $^{234,238}$U, $^{230,232}$Th) for the cryoconite samples are presented in Table S3 (Supplementary material). Activity concentrations range from 490±60 to 12300±600 Bq kg$^{-1}$ for total $^{210}$Pb, 12±1 to 43±4

Bq kg$^{-1}$ for $^{234}$U, 10±1 to 37±4 Bq kg$^{-1}$ for $^{238}$U, 16±1 to 54±4 Bq kg$^{-1}$ for $^{230}$Th and 16±1 to 58±4 Bq kg$^{-1}$ for $^{232}$Th. The highest activity concentration for $^{210}$Pb was observed in sample 4. Activity ratio of $^{234}$U/$^{238}$U varied between 0.9±0.1 to 1.4±0.3 (Tab. S5).

### 3.3 Heavy metals analyses in tundra soil profiles

The concentrations of all measured metals and ratios of stable Pb isotopes ($^{206}$Pb/$^{207}$Pb, $^{208}$Pb/$^{206}$Pb) in soil samples are presented in Tab. S6. Cd, Pb, Cu and Zn concentrations are shown in Figure 2C. Pb concentrations varied from 9.6 mg·kg$^{-1}$ (sample S05-3) to 25.9 mg·kg$^{-1}$ (sample S04-1). Cd concentrations range from 0.05 mg·kg$^{-1}$ (sample S01-2) to 0.6 mg·kg$^{-1}$





(sample S01-4). The concentrations of Zn ranged from 45.1 mg·kg$^{-1}$ (sample S03-1) to 76.9 mg·kg$^{-1}$ (sample S04-1). Cu concentrations varied from 11.8 mg·kg$^{-1}$ (sample S03-1) to 26.6 mg·kg$^{-1}$ (sample S01-2). The stable isotope ratio of $^{206}$Pb/$^{207}$Pb ranged from 1.190 to 1.217 while $^{208}$Pb/$^{206}$Pb ranged from 2.014 to 2.057. There was a significant correlation between Pb and Cu and Zn (r=0.82, r=0.71) (Tab. S7). Pb also correlated with naturally derived metals Ni, Cr and Co. Cu

correlated with Zn (r=0.77). Zn and Cu also correlated with Ni, Cr and Co. Interestingly, Cd did not correlate with any of these metals. The correlations between contaminating heavy metals and Fe and Al were checked to avoid the influence of natural variability of crust composition to metal concentration. Pb, Zn and Cu correlated with both metals, particularly strong was the correlation of Cu and Zn (r ranging from 0.73 to 0.84). The correlation of Pb with Fe and Al was much lower (r ranging from 0.53 to 0.59). $^{206}$Pb/$^{207}$Pb inversely correlates with Pb (r=0.78), lower but significant correlations were

measured for Cu, Zn and Cr. $^{208}$Pb/$^{206}$Pb did not correlate with any metal concentrations. The metal enrichment factors (normalized to Al) for soil samples ranged from 1.1 to 1.8 for Pb, from 0.9 to 1.6 for Zn, from 0.8 to 2.1 for Cu and from 0.4 to 7.7 for Cd (Tab. S6).

### 3.4 Heavy metals analyses in cryoconite samples

Table S8 presents concentrations of all measured metals and ratios of stable Pb isotopes ($^{206}$Pb/$^{207}$Pb, $^{208}$Pb/$^{206}$Pb) in cryoconite samples. Metals that are believed to be the most frequent contaminants (Cd, Pb, Zn, Cu) are also shown in Fig. 4. Pb concentrations ranged from 19.9 mg·kg$^{-1}$ (sample 13) to 97.7 mg·kg$^{-1}$ (sample 12). Cd concentrations varied from 0.2 mg·kg$^{-1}$ (sample 10) to 0.6 mg·kg$^{-1}$ (sample 11). Zn concentrations ranged from 59.6 mg·kg$^{-1}$ (sample 7) to 97.5 mg·kg$^{-1}$ (sample 11). Cu concentrations varied from 21.5 mg·kg$^{-1}$ (sample 13) to 40.1 mg·kg$^{-1}$ (sample 9). The ratio of $^{206}$Pb/$^{207}$Pb

ranged from 1.169 to 1.199 while $^{208}$Pb/$^{206}$Pb ranged from 2.025 to 2.061. There was a significant correlation between Pb and Cu (r=0.78) (Tab. S9). Pb also correlated with naturally derived metals Ni and Mn. Zn and Cu correlated with Cr and Ni. Interestingly, like in the soil samples, Cd did not correlate with any of the metals. The correlations between contaminating heavy metals and soil composition markers Fe and Al were checked to test the influence of natural variability of soil composition to metal concentration. There was no significant correlation of contaminating metals with Fe and Al, while

naturally derived Cr, Ni and Co correlated with Fe and Al. $^{206}$Pb/$^{207}$Pb inversely correlated with Pb (r=0.86), lower but significant correlations were measured for Cu, Ni and Mn. $^{208}$Pb/$^{206}$Pb correlated with only Pb and Cu (r=-0.58 and r=-0.60 respectively). The metal enrichment factors (normalized to Fe) for cryoconite samples ranged from 1.5 to 6.1 for Pb, from 1.0 to 1.5 for Zn, from 1.3 to 2.3 for Cu and from 0.5 to 4.5 for Cd (Tab. S8).



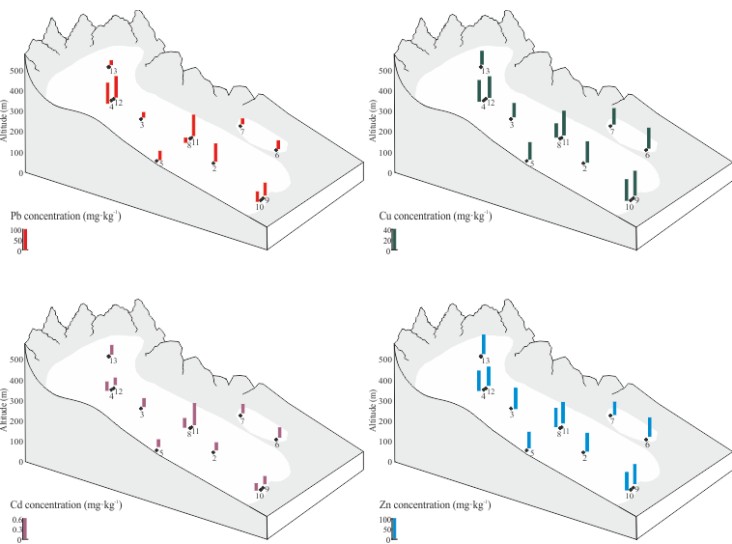

Figure 4. Concentrations of all measured metals (Cu, Pb, Cd, Zn) in cryoconite samples.

For most metals there were significant differences in concentration observed in the soil versus the cryoconite samples (Tab.
S10). The only exception is Cd which was similar (and elevated) in both environments. There was no difference in
$^{208}Pb/^{206}Pb$ ratio between soil and cryoconite samples but this is not very surprising since this ratio is less specific than
$^{206}Pb/^{207}Pb$ (the $^{208}Pb/^{206}Pb$ coal burning signature has a similar ratio to a natural rock).

## 4 Discussion

### 4.1 Radionuclides

All soil profiles show typical activity concentrations and inventories as do other profiles found on the raised marine terraces
covered by dry tundra in the western part of Spitsbergen (Łokas et al., 2014; 2017a,b; AMAP, 2015; Dowdall et al., 2003;
Gwynn et al., 2004). Retention of airborne radionuclides in tundra soils is facilitated by the presence of the relatively well-
developed organic horizon. The average values of the radionuclide inventories fall within the ranges of the above-mentioned
regional evaluations of fallout deposition (Hardy et al., 1973; Holm et al., 1983; AMAP, 2015; Łokas et al., 2017a,b).

Cryoconite materials are highly enriched with radionuclides of atmospheric origin compared with tundra soils. Radionuclide
enrichment exceeding the soil levels can be explained to some extent by the focusing of fine-grained material, washed down
into very small depressions where the cryoconite samples were collected, and also by the properties of cryoconite material.
Cryoconite granules are the mixtures of mineral particles, organic matter and microorganisms. They can retain and
concentrate airborne radionuclides due to metal-binding properties of extracellular substances that are excreted by
microorganisms. Differences in activity concentrations of anthropogenic radionuclides ($^{137}Cs$, Pu isotopes and $^{241}Am$) found
among the cryoconite samples reflect the different locations of sampling sites, morphological features and contents of
organic material. A significant correlation was observed between the amount of organic matter present and the concentration



of all airborne radionuclides. We noticed significant correlation between $^{210}$Pb, $^{137}$Cs, Pu isotopes and $^{241}$Am with the altitude and the area of the cryoconite holes. The depth of these cryoconite holes also correlates with $^{210}$Pb and Pu isotopes (Tab. S8). Cryoconite samples collected at the highest location (4, 12, 13) with the highest surface area of cryoconite holes (Tab. S1) were characterised by the highest observed activity concentrations. These three points were located very close to each other and the deposited material could have originated from the same source and migrated with supraglacial streams. In contrast, sampling points 5, 6, 7 which were located close to the moraine, were characterised by the lowest activity concentrations of anthropogenic radionuclides. These materials could have been washed out, mixed with mineral particles from the moraine or collapsed cryoconite holes (that may be destroyed more easily here than in the middle of the glacier).

The highest activity concentrations of $^{210}$Pb was observed in sample 4 (more than 12000 Bq kg$^{-1}$). More uniform and comparable values (about 5000 Bq kg$^{-1}$) were found for cryoconite samples 3, 8, 11, 12, 13. This $^{210}$Pb distribution is very similar to the distribution of anthropogenic radionuclides. Cryoconite granules can be locally redistributed on the surface of ice, dependent on the glacier morphology and altitude. It was noticed that activity concentrations increase in relation to the altitude above sea level. Material deposited in the highest locations can migrate with supraglacial streams and can be transported along with the glacier. Samples 4 and 13 were collected from typical cryoconite holes with the highest surface area (476 and 1200 cm$^2$) (Tab. S1). Regardless of the lifespan of individual cryoconite holes, their collapse does not imply removal of cryoconite from glacier surface as the dispersed cryoconite granules initiate formation of new holes (Takeuchi et al., 2001).

The constant delivery of $^{210}$Pb from the atmosphere suggests that the concentration of this radionuclide in cryoconite material should be proportional to the exposure time, while high concentrations of the artificial radionuclides indicate significant contribution of material that was exposed to the stratospheric or tropospheric fallout. The results show that $^{210}$Pb is being deposited at a more constant rate and lower values for samples 3, 8, 11, 12, 13 may suggest that the majority of material in these samples was already removed with melt waters or that material in these samples is younger than in sample 4 (shorter contact time with the atmosphere).

The effect of glacial morphology on effective trapping and storing of radionuclides was also observed in Georgia glacier (Łokas et al., 2018). The differences in the concentrations of radionuclides between sampling points and the lack of clear differences in the elevation gradient from terminus towards icefall may reflect the heterogeneous topography of the glacier tongue. In the Waldemarbreen we observed differences in the elevation gradient from the top of ice.

Activity concentrations of $^{234,238}$U and $^{230,232}$Th show little variability within the profiles and do not differ from values reported for soils globally (UNSCEAR, 1993), in the tundra soils in northern Spitsbergen (Dowdall et al., 2003) and in south-western Spitsbergen (Łokas et al., 2017b). Soil activity concentrations of these lithogenic radionuclides are related to their contents in the source minerals (Megumi et al., 1988) and are not influenced by the action of cryoconite. The presence of uranium and thorium are the main elements contributing to natural terrestrial radioactivity. Uranium isotopes ($^{234}$U and $^{238}$U) in terrestrial samples (rocks, soils and sediments) are usually present in radioactive equilibrium. The main source of





uranium in the natural environment is the atmospheric precipitation of terrigenic material, soil resuspension and rock weathering.

Our data for uranium analysis in cryoconite samples are comparable to other results for these materials. Our previous paper (Łokas et al., 2018) showed similar results for the concentration of $^{234}$U and $^{238}$U (mean value 33±3 and 34±3 Bq kg$^{-1}$,

respectively) in cryoconite granules from the Georgia glacier, while on the Swiss glacier (Baccolo et al., 2017), they reach around 118 Bq kg$^{-1}$ (with the mean value for $^{238}$U 65±5 Bq kg$^{-1}$). The value of the $^{234}$U/$^{238}$U activity ratio in analysed cryoconite granules suggests a state of radioactive equilibrium. The activity concentrations of $^{232}$Th are also in agreement with cryoconite samples from the Georgia glacier (mean value 42±4 Bq kg$^{-1}$) but higher mean values (65±3 Bq kg$^{-1}$) were observed in Swiss glaciers. The mean value for $^{230}$Th in analysed cryoconite samples is lower than for the Georgia glacier

(51±4 Bq kg$^{-1}$). The activity ratio between thorium isotopes ($^{230}$Th/$^{232}$Th) in cryoconite does not show a constant value – it varies from 0.4 to 0.7. In only a few samples the activity ratio of $^{230}$Th/$^{232}$Th is about 1 (samples 2, 3, 4, 12, 13). Isotopes derived from uranium ($^{230}$Th) and thorium series ($^{232}$Th) are not correlated in cryoconite samples ($R^2$=0.13) but in tundra soil profiles are better correlated ($R^2$=0.62). This may suggest that thorium from the uranium series is removed with melt water from the cryoconite samples located at the top of the glacier. We also observed differences between activity concentrations

of uranium and thorium isotopes in analysed tundra soils and cryoconite samples. In tundra soils the values are lower than in cryoconite.

## 4.2 The source of radionuclides and heavy metals contamination

Activity ratios of $^{238}$Pu/$^{239+240}$Pu, $^{239+240}$Pu/$^{137}$Cs $^{241}$Am/$^{239+240}$Pu and $^{240}$Pu/$^{239}$Pu atom ratio are commonly used to identify and

distinguish between global (stratospheric) and regional (tropospheric) sources of these radionuclides (Oughton et al., 2004; Hirose and Povinec, 2015; Łokas et al., 2013, 2014, 2016 and 2017b).

The $^{238}$Pu/$^{239+240}$Pu activity ratios in cryoconite samples are consistently higher than for tundra soils, suggesting the presence of contributing sources other than the global Pu fallout in this region. The percentage of global fallout versus local sources was estimated and the model assumes that the $^{238}$Pu/$^{239+240}$Pu activity ratios for global fallout and Chernobyl plutonium are

about 0.025 and 0.24, respectively. Following formula described by Mietelski and Wąs (1997) was used:

$$f_G = (A_R - A_M)/(A_R - A_G) \qquad (4)$$

where $f_G$ represents the fraction of plutonium isotopes from global fallout sources (%), $A_R$ represents the $^{238}$Pu/$^{239+240}$Pu ratio of Chernobyl sources (0.24), $A_M$ represents the measured $^{238}$Pu/$^{239+240}$Pu activity ratio in cryoconite samples and $A_G$ represents the $^{238}$Pu/$^{239+240}$Pu activity ratio of global fallout (0.025).

Calculations were performed for each of the studied cryoconite samples, and the fraction of global fallout varied from 82±1 % to 97±1 % (Fig. 5). Samples 4, 8 and 12, with the highest activity ratios of $^{238}$Pu/$^{239+240}$Pu (0.062), contained also the highest contribution of Pu from other sources (18 % and 17 %, respectively). The highest activity ratios of $^{238}$Pu/$^{239+240}$Pu (0.075 to 0.09) were observed in the initial soils from the vicinity of the Werenskiold glacier terminus in south-west part of




Spitsbergen. These values fall within the range found also for cryoconite from the Hans glacier (between 0.037 and 0.118 with a mean 0.064, Łokas et al., 2016). The $^{240}$Pu/$^{239}$Pu atomic ratios in analysed tundra soils (average of 0.179) (Fig. 6A) are comparable to global fallout ratios (0.183 ± 0.009) reported by Efurd et al. (2005) at 70° N latitude and 0.180 reported by others (Muramatsu et al., 1999; Warneke et al., 2002). Similar results was also reported for tundra soils from Spitsbergen (80 samples) with an average value of 0.179 (Łokas et al., 2017a). The tundra sites exposed to the atmosphere over the whole period of anthropogenic radionuclide release acquired their radionuclide contents primarily from direct atmospheric deposition.

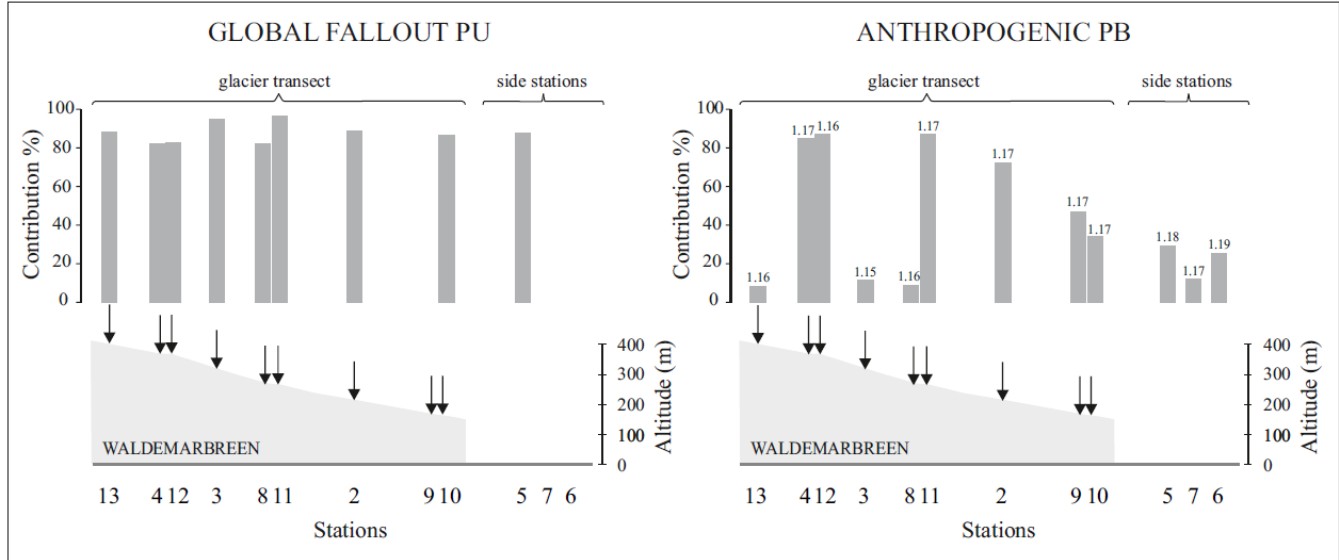

Figure 5. The contribution of global fallout Pu calculated using plutonium isotope ratios (left) contribution of anthropogenic origin Pb calculated using Pb isotopes and end-member method.

In tundra soils, $^{240}$Pu/$^{239}$Pu atomic ratios and the $^{238}$Pu/$^{239+240}$Pu activity ratios are similar to global fallout signature (0.180 and 0.025, respectively) but in cryoconite samples they are different (Fig. 6B). Tundra soils develop very slowly; therefore, the $^{240}$Pu/$^{239}$Pu atomic ratios are more uniform and homogeneous than in cryoconite samples. Soil profile have yearly stratification and averaged Pu isotopic ratios, resulting in small variation of the measured values. In contrast, in cryoconite samples, each such "snapshot" contains Pu, which was previously in a certain site in glacial and came from fallout after a specific deposition event. Similar mechanism was observed in initial soils from the proglacial zones of glacier from Spitsbergen (Łokas et al., 2017a). In the analysed cryoconite samples an average $^{240}$Pu/$^{239}$Pu atomic ratio is 0.147 (Fig. 6A) and this value is lower than for initial soils and also lower than in cryoconite from the Georgia glacier (0.163) (Łokas et al., 2018). The result for analysed cryoconite is very similar to initial soils and we checked the correlation between excess $^{238}$Pu and $^{240}$Pu/$^{239}$Pu atomic ratios (Figure 6C) and the lack of this indicates that enrichment of $^{238}$Pu is derived from sources other than $^{239}$Pu. We assumed in a previous article that the main source of Pu in the proglacial soils and now also in cryoconite



samples are nuclear tests and a non-exploded weapons-grade material. A third potential source of Pu is pure $^{238}$Pu from a satellite re-entry after injection from the stratosphere into the troposphere.

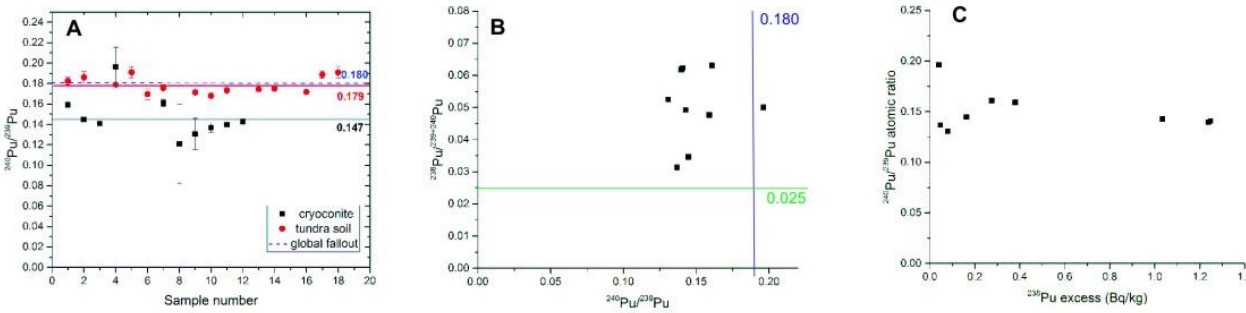

Figure 6. The relation between the $^{240}$Pu/$^{239}$Pu atomic ratio and sample number in tundra soils and cryoconite samples (A); the relation between the $^{240}$Pu/$^{239}$Pu atomic ratios and the $^{238}$Pu/$^{239+240}$Pu activity ratios in tundra soils and cryoconite samples (B); the correlation between excess $^{238}$Pu and $^{240}$Pu/$^{239}$Pu atom ratios in cryoconite samples (C).

The average $^{239+240}$Pu/$^{137}$Cs ratios in tundra soils (0.042) is similar to value of ~0.039 in the global fallout expected for the year 2013 (UNSCEAR, 1982) but in cryoconite samples (0.020) it is lower. The observed average value of about 0.021±0.03 in the proglacial zone of the glacier from west Spitsbergen is remarkable as there were no other such low values as at this site. Similar or slightly higher values (0.021 – 0.035) were observed on Svalbard by Gwynn et al. (2004) and Łokas et al. (2014). The enrichment of soils in $^{137}$Cs relative to plutonium might be due to the contributions from sources other than global fallout or these ratios may actually reflect the deposition ratios at this latitude. The mean $^{241}$Am/$^{239+240}$Pu ratios for tundra soils (0.47) and cryoconite samples (0.46) fall within the ranges reported for soils from western Spitsbergen (Łokas et al., 2014; Łokas et al., 2013) and other Arctic sites (Holm et al., 1983; Smith et al., 1997; Gwynn et al., 2005).

### 4.3 Heavy metals

The metal concentrations in soils were typical or slightly higher than concentrations characteristic for natural background concentrations. This had been confirmed by generally low metal enrichment ratios (EF ranging from 0.8 to 2.0). The only exception was Cd reaching up to 0.6 mg·kg$^{-1}$ (EF=7.7). The metal distributions measured in this study are very similar to metal concentrations characteristic for parent rocks (natural crust) on Svalbard. For instance, parent rocks in Hornsund contain very low concentrations of heavy metals (Pb~11 mg·kg$^{-1}$, Cd~0.04 mg·kg$^{-1}$, Zn~58 mg·kg$^{-1}$, Cu~20 mg·kg$^{-1}$; Samecka-Cymerman et al., 2011). Visible enrichment in Cd and Pb was found in surface and subsurface soil profile layers (Fig. 6). Similar enrichment in metal concentrations (Pb up to 25.3 mg·kg$^{-1}$, Cd up to 0.4 mg·kg$^{-1}$) in soil profiles of Kaffiøyra was found in the 1980's by Plichta (1993). Generally low metal concentrations have been mapped by Ottesen et al. (2010) in alluvial soils in Kaffiøyra region. Alluvial sediments are believed to represent natural metal levels, although recently deposited layers can be polluted by anthropogenic activity. Cd concentrations ranged from 0.01 to 0.13 mg·kg$^{-1}$ (our



results show values 0.05-0.6 mg·kg$^{-1}$), Cu ranged from 13.4 to 43.4 mg·kg$^{-1}$ (our values 11.8-26.6 mg·kg$^{-1}$), Pb ranged from 8.9 to 24.3 mg·kg$^{-1}$ (our values 9.6-25.9 mg·kg$^{-1}$) and Zn 58.9 to 79.2 mg·kg$^{-1}$ (our values 45.1-76.9 mg·kg$^{-1}$). Metal concentrations were studied in peat cores collected from nearby Kongsfjorden. The Cu ranged from 7 to 25 mg·kg$^{-1}$, Pb from 10 to 35 mg·kg$^{-1}$ while Zn ranged from 20 to 75 mg·kg$^{-1}$ (Headley, 1996). The region of Kongsfjorden may be have been

impacted by former mining activity, however (Hao et al., 2013). More recently, the metal distribution in soils of other Svalbard regions (Kongsfjorden and Adventfjorden) was studied by Halbach et al. (2017). Like our study, they found elevated concentrations of some metals eg. Cd (0.13-1.00 mg·kg$^{-1}$) in surface soils (covered by moss), but low concentrations of metals (Cd=0.04-0.5, Cu=3.3-28.4 mg·kg$^{-1}$, Pb=2.9-22.7 mg·kg$^{-1}$ and Zn=25-106 mg·kg$^{-1}$) in mineral soils (sampled at 20cm depth). Slightly higher levels of heavy metals (Cu: 17-92 mg·kg$^{-1}$ and Cd: 0.05-1.20 mg·kg$^{-1}$) were measured in

biologically rich soils in Hornsund region with the highest values in areas of rich vegetation (Wojtuń et al., 2013). Ziółek et al. (2017) reports much higher metal concentrations in peat soils in Bellsund. In their study Pb reached 100 mg·kg$^{-1}$, Zn reached 140 mg·kg$^{-1}$, Cu 57 reached mg·kg$^{-1}$ while Cd reached 8 mg·kg$^{-1}$. The high contamination of peat soils was in their case caused by the richness of organic matter due to a bird colony. In conclusion, the effect of anthropogenic influence on soils in Kaffiøyra region was very limited. This is also confirmed by relatively high $^{206}$Pb/$^{207}$Pb ratio (1.190 - 1.217)

measured in Kaffiøyra soils that is close to the natural ratio for parent rocks (>1.215) or/and represents the mixture of natural and anthropogenic Pb sources (Zaborska et al., 2017).

While soils can receive metals from atmosphere and from the surface run-off, cryoconites are fed only by atmospheric particles. Thus, interestingly, metal concentrations in cryoconite were higher than in soil samples (Tab. S8) and definitely exceeded natural values. Particularly elevated was the concentration of Pb reaching 97.5 mg·kg$^{-1}$ (EF = 6.1) and

Cd reaching 0.6 mg·kg$^{-1}$ (EF=4.5). The enrichment in other metals was lower (Zn up to 97.5 mg·kg$^{-1}$ and Cu up to 40.1 mg·kg$^{-1}$). The maximum concentration of Pb in cryoconite samples from the Hansbreen glacier in Hornsund was lower (82.7 mg·kg$^{-1}$), the concentrations of Zn (108 mg·kg$^{-1}$) and Cd (1.53 mg·kg$^{-1}$) were higher, however (Łokas et al., 2016). Singh et al. (2013) found a similar range of metal concentrations in cryoconite samples from a glacier located also in Kongsfjorden area (Pb up to 85.1 mg·kg$^{-1}$, Cu up to 44 mg·kg$^{-1}$, Cd up to 0.14 mg·kg$^{-1}$, Zn up to 150 mg·kg$^{-1}$). Chmiel et al. (2009) did not

find elevated metal levels in glacier waters in the Scott Glacier in Bellsund, they found enrichment in some metals (Pb = 190 mg·kg$^{-1}$) in cryogenic sediments, however. These reports suggest that the metallic trace elements, which are transported in the atmosphere attached to airborne particulate matter, tend to be retained and concentrated in the cryoconite material. This finding is confirmed by relatively lower $^{206}$Pb/$^{207}$Pb ratios in cryoconite samples (1.169 to 1.199). These ratios suggested that Pb in cryoconite is a mixture of natural and anthropogenic origin and the Pb of anthropogenic origin prevails. Using the end-

member method and based on stable isotopic Pb ratios, the contribution of anthropogenically derived Pb was estimated in cryoconite. It was found that Pb with an anthropogenic origin constituted 10-100 % of Pb accumulated in cryoconite samples (Fig. 5). In the case of the Horsund site (Hornbreen) the anthropogenic lead fraction in cryoconite was lower and ranged from 29 % to 95 % (Łokas et al., 2016). It was also calculated that the excess Pb in cryoconite material (Pb$_{measured}$ − Pb$_{natural}$), that ranged from 7.9 mg·kg$^{-1}$ to 85.7 mg·kg$^{-1}$, was characterized by $^{206}$Pb/$^{207}$Pb of 1.156-1.189 (Fig. 5). Side stations located





closer to rocky glacier sides were characterized by higher (more natural) computed anthropogenic Pb isotopic ratios (1.17-1.19) as compared to stations located in the central transect of the glacier (1.16-1.17). Since glaciers are fed by airborne particles cryoconite $^{206}Pb/^{207}Pb$ ratios can be compared to aerosol $^{206}Pb/^{207}Pb$ ratios. The study of aerosols in Kongsfjorden region recognized the dominance of two sources of Pb to that region. In the spring, isotopic signatures of $^{206}Pb/^{207}Pb$ ~1.158

suggests an Eastern Eurasian source, while in the summer a signature of $^{206}Pb/^{207}Pb$ ~1.167 suggests a mixture of North American and Eurasian influence (Bazzano et al., 2015). Thus, the cryoconites $^{206}Pb/^{207}Pb$ ratios compare well with aerosol $^{206}Pb/^{207}Pb$ ratios particularly in the center part of the glacier. This implies that it is mostly anthropogenic and not natural Pb that accumulates in the cryoconite samples and it can be said that an anthropogenic environmental impact is clearly visible in the Svalbard cryosphere.

Both radionuclides and heavy metals are deposited from the atmosphere on land/ocean and glacier surfaces by wet and dry precipitation. Since there are no new sources of radioactive pollution into the atmosphere, the patterns of atmospheric circulation do not influence radionuclide distribution within different glaciers on Svalbard. The amount of precipitation received by the glacier can definitely influence the glacier pollution, since coastal areas (eg. Kaffiøyra) receive twice the amount of precipitation compared to the inland areas (Førland et al., 2011). Post-depositional processes, mainly

those connected to cryoconite hole formation and eventual material transport within the glacier, appear to be a more significant influence on the levels of radionuclide pollution within the glacier. The different scenario concerns the distribution of heavy metals. Since they are still emitted to the atmosphere from different sources, the direction of atmospheric circulation is an important factor shaping the metal pollution distribution within the glacier. The atmospheric circulation over the Arctic has been intensively studied for decades. It is believed that Svalbard receives air masses mainly

from Europe and Russia (Reimann and de Caritat, 2005; Isaksen et al., 2016). Air mass transport direction varies seasonally and is influenced by the Arctic Oscillation (AO), however. The largest transport of contaminants to the Arctic occurs in winter and early spring (so-called Arctic haze). In the winters characterized by a positive AO index airborne contaminants originating from Europe and eastern parts of North America are dominant; conversely, during winters characterized by a negative AO index contaminants of Eurasian origin prevail. In the summer, contaminants associated with air masses from

North America and Europe dominate as the main transport direction (Macdonald et al., 2005). Local winds can also enhance the introduction of contaminants to coastal areas due to transport of contaminated sea salt aerosol (Lüdke et al., 2005 Kozak et al., 2015). The Kaffiøyra land area is shielded from winds from the seaward direction by the mountainous Prins Karl Forland island from the west. The dominating wind directions are north-west and south-east. Moreover, it was recently found that the winds became stronger (when compared to the period of 1975-2014) due to the intensification of cyclonic activity

(Przybylak et al., 2016). The recent intensification of wind activity may play an important role as a factor enhancing pollution of cryoconite granules. Unfortunately, due to lack of data in the previous year, the importance of atmospheric circulation variability could not be properly assessed.



## 5 Conclusions

The extent of variations of activity concentrations of airborne radionuclides and some trace metals (Pb and Cd) in cryoconite samples from the Waldemarbreen exceeds ranges observed in soils surrounding this glacier. The mechanisms responsible for accumulation of atmospheric contaminates are not very clear but we observed the effect of glacial morphology on effective trapping and storing of airborne radionuclides. The average mass ratios of $^{240}$Pu/$^{239}$Pu for tundra soils are comparable to the characteristic ratio for global fallout but for cryoconite samples (0.147) are much lower than for surrounding soils and also for initial soils from other west cost of Spitsbergen. We observed excess of $^{238}$Pu in cryoconite samples similar as we observed in other initial soils in Spitsbergen and this enrichment of $^{238}$Pu is derived from other sources than $^{239}$Pu. This is pure $^{238}$Pu from a satellite re-entry after injection from the stratosphere into the troposphere. Monitoring of cryoconite samples allows identification of Pu release events in the form of a fallout associated with each separate event. Therefore, cryoconite is an excellent candidate for atmospheric contamination monitoring.

The lithogenic radionuclide contents vary in much narrower ranges and their variability reflects changing proportions of weathering products derived from diverse parent rocks.

*Authors Contribution.* E.Ł designed the idea of this study and performed gamma and alpha analyses. A.C. prepared the calibration for gamma analyses. P.K. performed uranium analyses. I.S. collected the soil and cryoconite samples. P.G. carried out the purification of Nd(Pu)F$_3$ alpha-spectrometric sources in preparation for MC-ICP-MS measurements and language improvement. A. Milton carried out MC-ICP-MS analysis. E.Ł., A.Z. wrote the manuscript. E.Ł., A.Z., P.G. and I.S. supervised the research and contributed to the interpretation of the data. A.M. make language correction.

*Acknowledgment.* The research was founded by grant 2016/21/B/ST10/02327 from the Polish National Science Center (NCN) (PI: Edyta Łokas)  and IFJ PAS statutory research resources.

*Competing Interests.* The authors declare that they have no competing interests (both financial or non-financial).

*Supplementary information.* Full data about environmental radioactivity are available in the Supplementary Material.

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
