# Peer review of "Airborne radionuclides and heavy metals in High Arctic terrestrial environment as the indicators of sources and transfers of contamination"

_The Cryosphere, 2019_

## Referee Comment (RC1) · Anonymous Referee #1 · 10 Apr 2019

General comments

This is an interesting paper, the theme that is discussed is original and relatively little investigated by the glaciology community. I don't have any concern about the methodology and about the presented data and results. It is clear that the authors have expertise in the field of environmental radioactivity and inorganic pollution. Despite these points I cannot support the publication of this paper in its current form. Now the paper would be suitable for publication in a journal specialized on radioactivity, not in a journal whose audience comes from many and diverse fields of science, as is the case with Cryosphere. The manuscript needs a deep language and structural revision. I tried to

improve the first part of the paper from this perspective, but I realized that to really fix this side of the manuscript a big effort is needed. This is a job for the authors. Given the importance of this point, I feel that a true and accurate evaluation of the paper will be possible only after this first shape revision. The readability of the paper is not good, there are several parts where it is difficult to follow the flow and the many data seem not well connected, also because the interpretation of the results is a little bit poor and the paper is unbalanced: the presentation of raw data is very long and detailed (even too much in my opinion), while their interpretation and discussion are poor. There is a sort of gap between what someone is expecting to find in the paper given its title and the actual content. Authors should avoid presenting and discussing extensively on numerical values and ratios in the main text, there are figures to this aim that are more useful and informative, in addition the authors could consider adding or changing some figures with new ones, where also literature data are shown, so as to strengthen the hypothesis presented in this paper. For example, the issue that Pu composition is compatible with an influence from satellite reentry, is something of new or relatively rare, or is it common to see such results? This is not clear because the data are only compared to global fallout and very few other cryoconite data, but the findings are not sufficiently highlighted. Would it be possible to extend the comparison (both graphically and in the main text), so as to evidence if what was found by the authors was a local signal or if it is more common? Discussion and interpretation in this sense should be expanded and improved, so as to allow a full comprehension of the paper also to people not well concerned about environmental radioactivity. Another suggestion is to shorten the manuscript, now it is very long and the impression during the reading is that the same could have been said using less and less words. For example, I wouldn't always threat as separate, soils, cryoconite, radionuclides and heavy metals. Discussing about them together is more difficult for the authors, but it is for sure good from the perspective of the reader, who would better appreciate the importance and the novelty of this paper.

The abstract should be adjusted, in its current shape it is not very informative and the many points that are touched in it sound a little bit as disconnected. Given the

importance of this section it would be desirable to rewrite it. Now it is a brief summary of the entire paper, but the reader misses the main conceptual points of the paper, that in my opinion are: cryoconite is a better absorber than soil, local glacier morphology plays an important role in determining the accumulatio of pollutants (this is one of the most interesting points from the glaciological perspective), probably cryoconite age also influences the process, in addition cryoconite seems capable of recording both global and more local events.

Something similar is for the introduction. Also this part is not easy to read, the authors should deeply revise these sections, improving the language and the general structure of the text, maybe cutting some sections that sounds too technical for people who is not in the field of environmental radioactivity.

One of the most difficult sections to follow is the results one. I suggest to the authors to merge the results and the discussion ones in one single paragraph. Now the results part consists in a presentation of number and concentrations that is not very informative. The same information is found in the supplementary tables. I see two options: 1-results are removed and its content is added to the following section, creating a single results and discussion paragraph. 2-the results part is strongly shortened and supported by figures that present the data in a graphical way (current figures are almost impossible to read).

Figures must be modified, in many cases font size and details are too little. Some figures are also too dark. To make them well readable, modifications are needed.

I suggest to the authors to improve figures, possibly adding some comparison between the samples considered here and ones discussed in previous publications, in particular dealing with isotopic ratios. It would be nice to understand if what is found in cryoconite is a common signal (even if amplified) or it is something of peculiar.

This paper could be potentially published in TC, but several additional efforts must be pushed by the authors to this aim, therefore major revisions is my final response.

Below more specific comments.

Page1

Line20: why weathered? I think that this term here is not fully appropriate, you are talking about local material, regardless of its weathering degree. I suggest to the authors to change the term with "local"

Line23: I would change to "from additional and more specific sources might be...". I would like to highlight the differences between a global and uniform fallout and the more local signals that seem to be recoded in cryoconite.

Line24: change "is visible" with "was detected"; change "are" with "is"

Line25-28: this statement sounds too much confident and assume many concepts as known by the reader. I would suggest changing with "Approximately one third of the total observed Pu activity concentration is related to 238Pu and can be explained considering the atmospheric re-entry of the SNAP9A satellite, which was powered by a Pu thermoelectric generator."; change with "In the sample from the Waldemarbreen glacier we could appreciate the influence of glacial local morphology on the capability of cryoconite of trapping and accumulating airborne radionuclides." Line 28-30: this passage is not clear, it seems that the sampling position on this glacier has an influence on the airborne radionuclide content of cryoconite. But the authors say that: The difference in the concentrations... may reflect the homogenous topography of the glacier tongue. How can a homogenous factor be related to a variable feature? The passage must be rephrased.

Line2: replace "elements" with "species" Line3: it seems that only the cited species are the artificial radionuclides, please adjust this passage saying that many and many radioactive nuclides were released in the environment by humans, but that the present work is focused on the cited ones.

Line7: "disintegration of satellites powered by nuclear thermo-electric generators"

Line11: what is the non-proliferation of nuclear material? Please better describe, the readers of cryosphere are usually not very confident with these themes.

Line15: this is not true. It seems that the cryosphere is only affected by the pollution from radionuclides and heavy metals, but these are only two examples. It is widely known that many and many different species related to human activities are currently found in the cryosphere. Please reformulate, spreading the message that metals and radioactivity are only two of the several pollutant groups that affect the cryosphere.

Line27: please specify where cryoconite is found also. It is important to say that cryoconite is only found on the ablation surface of glaciers.

Line29-30: this is an important passage, please add a reference for it.

Line31: change "their" with "its".

Line34: "Cryoconite is usually found. . ."

Line1: "Such holes are usually. . ."

Line3-4: this information is not correct. The value of 80% is referred to "Low Latitudes, Caucasus, Central Europe, Western Canada and US and Scandinavia" as it is originally stated in the cited paper (Radic et al., 2014). This is not a global value, since the response of Antarctica and Greenland will be completely different with respect to the other local and continental glacial systems. I suggest the reader to reformulate the sentence after having revised the cited paper.

Line5: "contain"

Line8-9: "provide evidence that cryoconite-derived contaminants were dispersed in the pro-glacial area by meltwater channels and accumulated in depressions of the glacier

forefield."

Line10: which properties? Maybe it will be better to use features, but anyway it is not clear what the authors are talking about.

Line12-14: it is clear what the authors are saying, but it would be nice to reformulate this passage. The topics touched here are quite sensitive and I would suggest not to directly cite specific countries.

Line25: the figure is too dark, and the used font is too little, it should be modified, now it is not possible to appreciate all the details.

Page5

Line26: introduce the coefficient r, what is it?

Page6

Line2: change "understood" with "defined"

Line10: change with "this is the reason why the activity…"

Line1: the authors should modify the figure, increasing the font size and the boxes dedicated to the vertical profiles, now you can't read anything.

Line18-20: I guess that this is only a hypothesis, even if very likely. The authors should be more open to doubt and add a reference.

Line1-2: please add the correlation coefficients to better appreciate the correlation degree between these variables.

Line5: "which were located close to the moraine at lower altitude"

Line8: please explain why here the collapse of cryoconite holes is more likely

Line8: what is mineral soil?

Line 19-30: this part sounds like an introduction, it should be removed (or drastically shortened) or moved to the introduction.
* * *

---

## Referee Comment (RC2) · Anonymous Referee #2 · 23 Apr 2019

Comment on "Airborne radionuclides and heavy metals in High Arctic terrestrial environment as the indicators of sources and transfers of contamination", E. Łokas et al.

Anonymous reviewer no. 2

General comments:

From an analytical perspective, this article represents a great amount of work with regard to sample collection and processing, chemical and radiochemical separations, instrumental measurements, and interpretation of results. A systematic organization of the measurement results is found in the extensive tables of analytical data; these will

enable valuable comparisons with past and future data obtained by other researchers in that region. Perhaps referring to the tables, figures and plots with very brief descriptions should comprise the "Results" section. I'm not sure that it is necessary to mention the ranges of all the analytes in the text when this information can be seen in the tables.

The "Discussion" section is partitioned into three subsections. Subsection 4.2 "The source of radionuclides and heavy metals contamination" is essentially entirely devoted to radionuclides; perhaps it would be better just to drop the "heavy metals" part from this title since it is discussed in depth in subsection 4.3.

Some of the actual discussion of the results, especially for stable heavy metals and correlations among them (subsections 3.3 and 3.4), occurs in the "Results" section. It is recommended to integrate these remarks into the "Discussion" section along with the existing material.

Uncertainties are presented with the tables S2 through S5 giving radionuclide measurement results; however, I cannot find any mention in the text of whether these uncertainties are at the 68% (k=1) or 95% (k=2) confidence level. A general statement about the confidence level of uncertainties should be given early in the text (e.g., at the very beginning of the "Results" section). In addition, the main contributions to the radioanalytical uncertainties should be mentioned (e.g., counting statistics, calibrations, tracer concentrations, etc.).

Table S6 lists all of the stable element measurement results, but there are no uncertainties attached to these values. A statement of uncertainty "ranges" associated with each element concentration measurement should be given for the reader, perhaps at the column headings or as a footnote.

My recommendation is that this article should be published with minor revisions as outlined above.

Specific comments

1. Although I believe Svalbard and Spitsbergen refer to the same archipelago, the authors should choose one or the other to be used consistently in the article.

2. In the abstract, it would be useful to mention that the Kaffioyra region is located in Spitsbergen/Svalbard.

3. In subsection 2.3.2 "Heavy metal analyses", it is not mentioned if separate sub-samples (from those used for radionuclide analyses) were used for the stable element determinations. If separate samples were used, how much material and what dissolution method(s) was (were) used to prepare them for the flame atomic absorption and mass spectrometer measurements?

4. How was sample organic content assessed and related to LOI (which needs to be defined in the text)?

5. The plots of depth distribution in Fig. 2 are important but very difficult to read; a significant improvement in quality is needed.

6. The reference "Łokas et al., 2010" in line 10 on page 5 is missing from the "References" section at the end of the article. I suspect it is Nukleonika 2010, volume 55(2), pages 195 – 199, but please correct this oversight.

7. In line 6 on page 17, "average mass ratios" should be "average atom ratios" to be consistent with all of the previous text.

8. In line 8 on page 17, ". . . initial soils from other west cost of Spitsbergen" is confusing; perhaps the authors meant ". . . initial soils from the other (west) coast of Spitsbergen"?

9. In lines 8-9 on page 17, perhaps to write "We observed an excess of 238Pu in cryoconite samples similar to that which we observed in other initial soils in Spitsbergen, and this enrichment of 238Pu is derived from sources other than 239Pu. This is due to pure 238Pu from a satellite re-entry . . ."

---

## Author Comment (AC1) · 7 Jun 2019

The Cryosphere Discussions, tc-2019-34 Title: Airborne radionuclides and heavy metals in High Arctic terrestrial environment as the indicators of sources and transfers of contamination

Dear Professor Flanner, Thank you very much for your handling of our revised manuscript (tc-2019-34). We have addressed, one by one, all comments made by the Reviewers. Below we provide our responses.

Anonymous Referee #1 Thank you for the thoughtful and constructive comments. We

[Figure]

agree with and have followed all comments from the reviewer 1 especially that the manuscript needed structural revision. The abstract and the introduction were revised, some parts were removed for the sake of clarity and ease of comprehension. The results part was strongly shortened and also revised. The raw data were mostly removed from the text.

General comments This is an interesting paper, the theme that is discussed is original and relatively little investigated by the glaciology community. I don't have any concern about the methodology and about the presented data and results. It is clear that the authors have expertise in the field of environmental radioactivity and inorganic pollution. Despite these points I cannot support the publication of this paper in its current form. Now the paper would be suitable for publication in a journal specialized on radioactivity, not in a journal whose audience comes from many and diverse fields of science, as is the case with Cryosphere. We feel that results of our paper are interesting for the broader audience of The Cryopshere because the artificial radioactivity that is assocciated with cryoconite appears to be a potential tracer of material transfers on and within glaciers and from glaciers to their forefronts. Radioactivity surveys on glaciers and their forefronts might become a usefull tool in the assessments of glacier responses to climate change. At the same time, the cryospheric community is mostly unaware of the phenomenon of fallout radioactivity (and other pollution) accumulation in cryoconite. Our results also indicate that the cryosphere is an important, and active, element of the global cycling of airborne pollution. Our research is also in line with the recent recognition of the role of biological processes on glacier surfaces in the cryospheric and global biogeochemical cycling.

The manuscript needs a deep language and structural revision. I tried to improve the first part of the paper from this perspective, but I realized that to really fix this side of the manuscript a big effort is needed. This is a job for the authors. Given the importance of this point, I feel that a true and accurate evaluation of the paper will be possible only after this first shape revision. The structure of the manuscript was revised. Some of

the subsections were merged. The language was improved by one of the co-authors (dr Andrew Milton).

The readability of the paper is not good, there are several parts where it is difficult to follow the flow and the many data seem not well connected, also because the interpretation of the results is a little bit poor and the paper is unbalanced: the presentation of raw data is very long and detailed (even too much in my opinion), while their interpretation and discussion are poor. There is a sort of gap between what someone is expecting to find in the paper given its title and the actual content. Authors should avoid presenting and discussing extensively on numerical values and ratios in the main text, there are figures to this aim that are more useful and informative, in addition the authors could consider adding or changing some figures with new ones, where also literature data are shown, so as to strengthen the hypothesis presented in this paper. For example, the issue that Pu composition is compatible with an influence from satellite re-entry, is something of new or relatively rare, or is it common to see such results? This is not clear because the data are only compared to global fallout and very few other cryoconite data, but the findings are not sufficiently highlighted. Would it be possible to extend the comparison (both graphically and in the main text), so as to evidence if what was found by the authors was a local signal or if it is more common? Discussion and interpretation in this sense should be expanded and improved, so as to allow a full comprehension of the paper also to people not well concerned about environmental radioactivity. Most of raw data was removed from the text. Fig. 5 presents additional data from literature (Łokas et al., 2018; Łokas et al., 2016 and our unpublished data from Arctic glaciers). The issue of 238Pu source from satellite re-entry is presented in the following added sentences: "A third potential source of Pu is pure 238Pu from a satellite re-entry after injection from the stratosphere into the troposphere. The 238Pu enrichment was observed also in other cryoconite from Arctic glaciers (unpublished data) (Fig. 5C) and in air filters in Finland (Salminen-Paatero et al., 2012) or in air filters from NW Poland (Kierepko et al., 2016). "

Another suggestion is to shorten the manuscript, now it is very long and the impression during the reading is that the same could have been said using less and less words. For example, I wouldn't always threat as separate, soils, cryoconite, radionuclides and heavy metals. Discussing about them together is more difficult for the authors, but it is for sure good from the perspective of the reader, who would better appreciate the importance and the novelty of this paper. The manuscript was shortened, also by removing most of raw data. The presentation of results for soils and cryoconite was merged. The results section is now divided into two subsections: "3.1. Radionuclide contents and their activity ratios in tundra soil profiles and in cryoconite samples" and "3.2. Heavy metals analyses in tundra soil profiles and in cryoconite samples". The discussion was also merged and divided only for 2 subsections: "4.1. Radionuclides and heavy metals contamination" and "4.2. The source of radionuclides and heavy metals contamination ".

The abstract should be adjusted, in its current shape it is not very informative and the many points that are touched in it sound a little bit as disconnected. Given the importance of this section it would be desirable to rewrite it. Now it is a brief summary of the entire paper, but the reader misses the main conceptual points of the paper, that in my opinion are: cryoconite is a better absorber than soil, local glacier morphology plays an important role in determining the accumulation of pollutants (this is one of the most interesting points from the glaciological perspective), probably cryoconite age also influences the process, in addition cryoconite seems capable of recording both global and more local events. The abstract was rewritten following reviewer's suggestions.

Something similar is for the introduction. Also this part is not easy to read, the authors should deeply revise these sections, improving the language and the general structure of the text, maybe cutting some sections that sounds too technical for people who is not in the field of environmental radioactivity. One of the most difficult sections to follow is the results one. I suggest to the authors to merge the results and the discussion ones in one single paragraph. Now the results part consists in a presentation of number

and concentrations that is not very informative. The same information is found in the supplementary tables. I see two options: 1-results are removed and its content is added to the following section, creating a single results and discussion paragraph. 2-the results part is strongly shortened and sup-ported by figures that present the data in a graphical way (current figures are almost impossible to read). The introduction was shortened and rewritten. The language was improved. The results and discussion sections were also corrected. In the result section we left only maximum value for radionuclides and only average values for isotopic ratios. For heavy metals we left ranges because Figure 3 presents only two: Pb and Cd. See replies to the above comments.

Figures must be modified, in many cases font size and details are too little. Some figures are also too dark. To make them well readable, modifications are needed. I suggest to the authors to improve figures, possibly adding some comparison between the samples considered here and ones discussed in previous publications, in particular dealing with isotopic ratios. It would be nice to understand if what is found in cryoconite is a common signal (even if amplified) or it is something of peculiar. This paper could be potentially published in TC, but several additional efforts must be pushed by the authors to this aim, therefore major revisions is my final response. All figures were modified. Fig 1 was lightened and we removed the Fig 1B with the recession of Waldemarbreen. Now the Fig 1 is better readable. The Figures 2 and 3 were merged (for Fig 2 corrected) and we selected only airborne radionuclides for cryoconite and soil profiles, we removed graphs with natural radionuclides like U and Th isotopes because these data are presented in the supplementary material and also we found higher differences in radioactivity between cryconite and soils for airborne radionuclides than for natural radionuclides. We also merged Figures 3 and 4 (now Fig 3). This new figure shows only two, most interesting heavy metals Cd and Pb in cryoconite and soil profiles. We removed other heavy metals from the graph because these data are presented in sup-plementary material. We also present additional data for cryoconite (some of that are not published yet) in the corrected Fig 5 (now Fig 6).

The presentation of raw data is very long and detailed, while their interpretation and discussion are poor. The authors could consider adding or changing some figures with new ones, where also literature data are shown The result was divided for two subsection: "Radionuclide contents and their activity ratios in tundra soil profiles and in cryoconite samples" and "Heavy metals analyses in tundra soil profiles and in cryoconite samples". Generally, this section was shortened, figures were merged and the selected airborne radionuclides in soils and cryoconite and selected heavy metals in soils and cryoconite and presented on Fig 2 and 3, respectively. Fig 1 was modified, it was lightened and larger font size was used. In Fig 5 we added some new literature data (Łokas et al., 2018; Łokas et al., 2016 and our unpublished data from Arctic glaciers).

Another suggestion is to shorten the manuscript, now is very long and the impression during the reading is that the same could have been said using less and less words eg. I wouldn't always threat as separate, soils, cryoconite, radionuclides and heavy metals. . . . The manuscript was shortened, the result for soils and cryoconite were merged. The results section was divided into two subsections: "Radionuclide contents and their activity ratios in tundra soil profiles and in cryoconite samples" and "Heavy metals analyses in tundra soil profiles and in cryoconite samples"

Abstract should be adjusted Abstract was adjusted as suggested by the reviewer.

Introduction should be revised The introduction was revised, some parts were removed for the sake of clarity

Results should be strongly shortened. . . Corrected

Figures must be modified, in many cases font size and details are too little. Some figures are also dark. Corrected. See above.

Specific comments

Page 1. Line 20: " . . . changed weathered with local" Corrected to: ". . ..their activity

concentrations are controlled only by mixing of local material derived from different bedrock."

Line 23: I would change to "from additional and more specific sources might be…" Corrected.

Line 24: change " is visible" with "was detected" and "are" with "is" Corrected to: "We assumed that the main source of Pu, which was detected only in cryoconite samples, is derived from nuclear tests and non-exploded weapons-grade material."

Line 25-28: change "Approximately one third of the total observed Pu activity concentration is related to 238Pu and can be explained considering re-entry of the SNAP9A satelite, which was powered by a Pu thermoelectric generator ." Changed to:: "Approximately one third of the total observed Pu activity concentration is 238Pu originating from a SNAP9A satellite re-entry, which was powered by a Pu thermoelectric generator ."

Line 25-28: this passage must be rephrased Replaced as:: "Local glacial morphology plays an important role for determining the accumulation of airborne pollutants."

Page 2. Line 2: replace "elements" with "species" Replaced

Line 3: ….adjust this passage It was adjusted: "Many of the artificial radionuclides were released into the environment due to various human activities mainly in the second half of the 20th century, but the present work is focused only on the 137Cs, 238,239,240Pu, 241Am."

Line 7: add: ".. powered by nuclear thermos-electric generators" Following part was added: "….disintegration of satellites (SNAP9A-1964, Cosmos 958-1978) powered by nuclear thermos-electric generators."

Line 11: what is the non-proliferation of nuclear material? Replaced with: "….radioactivity is closely connected with potential threats to national security and controlling of the spread of nuclear material."

Line 15: Please reformulate, spreading the message that metals and radioactivity are only two of the several pollutant groups that affect cryosphere Corrected to: "Additionally, the presence of these artificial radionuclides and heavy metals are only two of the several pollutant groups that affect the cryosphere."

Line 27: please specify where cryoconite is found also. It's important to say that cryoconite is only found on the ablation surface of glaciers It was reformulated: "…such holes are form on the ablation zone of glaciers and constitute nutrient-rich pools with diverse biota within the supraglacial zone"

Line 29-30: this is an important passage, please add a reference for it It was added two references: "They produce extracellular polymeric substances whose adhesive properties enhance the accumulation of mineral dust and microorganisms (Gadd 2004; Francis 2007)."

Line 31: change "their" with "is" Corrected

Line 34: "Cryoconite is usually found…." Corrected

Page 3. Line 1: "Such holes are usually found…" Corrected to: "Such holes are usually found on the ablation zone of glaciers…" Line 3-4: This information is not correct. It was reformulated: "Glaciers in Canadian, Russian Arctic, Alaska, Antarctica and Greenland are projected to lose about 40 % of their volume by the end of 2100 but in Central Europe, low-latitude South America, Caucassus, North Asia, Western Canada and US are expected to lose about 80 % of their volume or disappear within decades at current climatic conditions (Radic et al., 2014)."

Line 5: " contain" Corrected

Line 8-9: " provide evidence that cryoconite-derived contaminants were dispersed in the proglacial area by meltwater channels and accumulated in depressions on the glacier forefield" Corrected

Line 10: which properties? May be it will be better to use features, but anyway it is not

clear what authors are talking about This sentence was removed.

Line 12-14: it is clear what authors are saying but it would be nice to reformulate this passage We agree with this suggestion that this topic are quite sensitive and we refused the countries.

Page 4. Line 25: the figure is too dark and the used font is too little This Fig 1 was modified.

Page 5. Line 26: introduce the coefficient r, what is it? The citation was added: "….(eg. Mukaka, 2012)"

Page 6. Line 2: change "understood" with "defined" It was corrected

Line 10: change with "this is the reason why the activity…" It was corrected

Page 7. Line 1: the authors should modify the figure, increasing the font size and the boxes dedicated to the vertical profiles, now you can't read anything It was modified and corrected: Figures 2 and 3 were merged and the author selected airborne radionuclides and heavy metals and presented on new Fig 2 and 3, respectively.

Page 10. Line 18-20: I guess that this is only a hypothesis, even if very likely. The authors should be more open to doubt and add a reference It was rephrased and references were added: "Probably, they can retain and concentrate airborne radionuclides due to metal-binding properties of extracellular substances that are excreted by microorganisms (Gadd 2004; Francis 2007)."

Page 11. Line 1-2: please add the correlation coefficient to better appreciate the correlation degree between these variables It was added: "A significant correlation was observed between the amount of organic matter present and the concentration of all airborne radionuclides (r=0.79-0.83). We noticed significant correlation between $210Pb$, $137Cs$, Pu isotopes and $241Am$ with the altitude (r=0.64-0.91) and the area of the cryoconite holes (r=0.58-0.63), except $210Pb$ (r=0.38). The depth of these cryoconite holes also correlates with $210Pb$ and Pu isotopes (r=0.60-0.68)"

Line 5: "which were located close to the moraine at lower altitude" Corrected

Line 8: please explain why here the collapse of cryoconite holes is more likely The part of this sentence was removed.

Page 15. Line 8: what is mineral soil It was corrected: "..sandy fraction of soils.."

Page 16. Line 19-30: this part sounds like introduction, it should be removed (or drastically shortened) or moved to the introduction It was shortened and modified. "Both radionuclides and heavy metals are deposited from the atmosphere on land/ocean and glacier surfaces by wet and dry precipitation. Since there are no new sources of radioactive pollution into the atmosphere, the patterns of atmospheric circulation do not influence radionuclide distribution within different glaciers on Svalbard. The amount of precipitation received by the glacier can definitely influence the glacier pollution, since coastal areas (eg. Kaffiøyra) receive twice the amount of precipitation compared to the inland areas (Førland et al., 2011). Post-depositional processes, mainly those connected to cryoconite hole formation and eventual material transport within the glacier, appear to have more significant influence on the levels of radionuclide pollution within the glacier. The different scenario concerns the distribution of heavy metals. Since they are still emitted to the atmosphere from different sources, the direction of atmospheric circulation is an important factor shaping the metal pollution distribution within the glacier. Main atmospheric metal discharge origins from Europe and Russia (Isaksen et al., 2016), in coastal regions local winds can enhance the introduction of contaminants due to transport of contaminated sea salt aerosols, however (Lüdke et al., 2005 Kozak et al., 2015). "

Please also note the supplement to this comment:
https://www.the-cryosphere-discuss.net/tc-2019-34/tc-2019-34-AC1-supplement.pdf

[Figure]

**Fig. 1.** Location of the study area. Profile S01-S06 were collected at increasing distances from the front of Waldemarbreen and cryoconite samples (2-13) from different altitudes of this glacier.

[Figure]

**Fig. 2.** The model of Waldemarbreen with the activity concentration of atmospheric radionuclides (210Pb, 137Cs, 239,240Pu, 241Am) in cryoconite and depth distribution of these radionuclides in tundra soils.

[Figure]

**Fig. 3.** Concentrations of selected heavy metals (Pb, Cd) in cryoconite samples and tundra soil profiles.

[Figure]

**Fig. 4.** The contribution of global fallout Pu calculated using plutonium isotope ratios (left) contribution of anthropogenic origin Pb calculated using Pb isotopes and end-member method.

[Figure]

**Fig. 5.** The relation between the 240Pu/239Pu atom ratios and the 238Pu/239+240Pu activity ratios in investigated cryoconite samples, other Arctic cryoconite (Łokas et al., 2018 and unpublished data) and cryoc

---

## Author Comment (AC2) · 7 Jun 2019

The Cryosphere Discussions, tc-2019-34 Title: Airborne radionuclides and heavy metals in High Arctic terrestrial environment as the indicators of sources and transfers of contamination

We are thankful for the thorough and constructive comments and remarks on our manuscript. The issues raised by the reviewer were taken into consideration and in the following paragraphs, we present our reply to each of them. Below we provide our responses.

[Figure]

Anonymous Referee #2: General comments: From an analytical perspective, this article represents a great amount of work with regard to sample collection and processing, chemical and radiochemical separations, instrumental measurements, and interpretation of results. A systematic organization of the measurement results is found in the extensive tables of analytical data; these will enable valuable comparisons with past and future data obtained by other researchers in that region. Perhaps referring to the tables, figures and plots with very brief descriptions should comprise the "Results" section. I'm not sure that it is necessary to mention the ranges of all the analytes in the text when this information can be seen in the tables. We agree with the reviewer that the results section should be modified. The results for soils and cryoconite were linked. The results section was divided for two subsections: "Radionuclide contents and their activity ratios in tundra soil profiles and in cryoconite samples" and "Heavy metals analyses in tundra soil profiles and in cryoconite samples". Generally, this section was shortened, figures were merged and the author selected airborne radionuclides in soils and cryoconites and only two heavy metals in soils and cryoconite and presented on Fig 2 and 3, respectively. We will present additional data on radionuclides in cryoconites (some of that are not published yet) in the corrected Fig 5 (now Fig 6). In the results section we left only maximum value for radionuclides and only average values for isotopic ratios. For heavy metals we left ranges because Figure 3 presents only concentrations of two metals: Pb and Cd. We removed other heavy metals from the graph because these data are presented in supplementary material.

The "Discussion" section is partitioned into three subsections. Subsection 4.2 "The source of radionuclides and heavy metals contamination" is essentially entirely devoted to radionuclides; perhaps it would be better just to drop the "heavy metals" part from this title since it is discussed in depth in subsection 4.3.Some of the actual discussion of the results, especially for stable heavy metals and correlations among them (subsections 3.3 and 3.4), occurs in the "Results" section. It is recommended to integrate these remarks into the "Discussion" section along with the existing material. The discussion was modified, it was merged and divided only for 2 subsections: "4.1. Radionuclides

and heavy metals contamination" and "4.2. The source of radionuclides and heavy metals contamination ". Most of the raw data of radionuclides and heavy metals in the discussion were removed.

Uncertainties are presented with the tables S2 through S5 giving radionuclide measurement results; however, I cannot find any mention in the text of whether these uncertainties are at the 68% (k=1) or 95% (k=2) confidence level. A general statement about the confidence level of uncertainties should be given early in the text (e.g., at the very beginning of the "Results" section). In addition, the main contributions to the radioanalytical uncertainties should be mentioned (e.g., counting statistics, calibrations, tracer concentrations, etc.). The information about uncertainties and confidence level was added to the Methods section: "The uncertainties of activity concentrations include the measurement uncertainties (counting statistics and efficiency)and are reports as 1 ïĄş counting statistics."

Table S6 lists all of the stable element measurement results, but there are no uncertainties attached to these values. A statement of uncertainty "ranges" associated with each element concentration measurement should be given for the reader, perhaps at the column headings or as a footnote. The information about uncertainties was added to the "Methods" section: "The quality control and assurance for this method is presented in Zaborska et al. (2017)." My recommendation is that this article should be published with minor revisions as outlined above.

Specific comments to revised article: 1. Although I believe Svalbard and Spitsbergen refer to the same archipelago, the authors should choose one or the other to be used consistently in the article. The authors choose Spitsbergen.

2. In the abstract, it would be useful to mention that Kaffioyra region is located in Spitsbergen "Spitsbergen" was added.

3. In subsection 2.3.2 "Heavy metal analyses" it is not mentioned if separate subsamples (from those used for radionuclide analyses) were used for stable element determinations. If separate samples were used, how much material and what dissolution methods were used to prepare them for the flame atomic absorption and mass spectrometer measurements? Following text was added: "Separate sub-samples (0.5 g) were dedicated for measurements of selected heavy metals that are believed to be important anthropogenic contaminants (Pb, Cu, Zn, Cd). Contaminating metals and other metals (Fe, Al, Ni, Cr, Mn) were measured with a flame atomic absorption spectrometer (AAS Shimadzu 6800) using deuterium background correction."

4. How was sample organic content assessed and related to LOI (which needs to be defined in the text)? It was defined in the "Methods" section: "Organic matter contents were determined by loss on ignition (LOI) at 600° C for 6 hours."

5. The plots of depth distribution in fig 2 are important but very difficult to read; a significant improvement in quality is needed. Figures 2 and 3 were merged and the authors selected airborne radionuclides and selected heavy metals and presented on new Fig 2 and 3.

6. The reference "Łokas et al., 2010" in line 10 on page 5 is missing. . . . . . . It was added to the "References" section.

7. In line 6 on page 17, " average mass ratios" should be "average atom ratios" to be consistent with all of the previous text. Corrected.

8. In line 8 on page 17, ". . . .initial soils from other west cost of Spistbergen" is confusing. . . . .. It was corrected as ". . . .initial soils from other west coast of Spitsbergen".

9. In line 8-9 on page 17, perhaps to write "We observed an excess of 238Pu in cryoconite samples similar to that which we observed in other initial soils in Spitsbergen, and this enrichment of 238Pu is derived from sources other than 239Pu. This is due to pure 238Pu from a satellite re-entry after injection from the stratosphere into the troposphere." Corrected.

Please also note the supplement to this comment:
https://www.the-cryosphere-discuss.net/tc-2019-34/tc-2019-34-AC2-supplement.pdf

[Figure]

**Fig. 1.** Location of the study area. Profile S01-S06 were collected at increasing distances from the front of Waldemarbreen and cryoconite samples (2-13) from different altitudes of this glacier.

[Figure]

**Fig. 2.** The model of Waldemarbreen with the activity concentration of atmospheric radionu-clides (210Pb, 137Cs, 239,240Pu, 241Am) in cryoconite and depth distribution of these ra-dionuclides in tundra soils.

[revised manuscript text omitted]